# Decomposition and Nutrient Release from *Larix olgensis* Stumps and Coarse Roots in Northeast China 33-Year Chronosequence Study

Xiuli Men [1], Yang Yue [1], Xiuwei Wang [1,2] and Xiangwei Chen [1,2,*]

[1] School of Forestry, Northeast Forestry University, Harbin 150040, China; menxiuli0807@nefu.edu.cn (X.M.); 2019023133@nefu.edu.cn (Y.Y.); wxgreat@nefu.edu.cn (X.W.)
[2] Key Laboratory of Sustainable Forest Ecosystem Management, Ministry of Education, Northeast Forestry University, Harbin 150000, China
[*] Correspondence: xwchen1966@nefu.edu.cn; Tel.: +86-13313623237

**Abstract:** Stumps and coarse roots form an important C pool and nutrient pool in a *Larix olgensis* (Larix olgensis Henry) plantation ecosystem, and their decomposition processes would affect nutrient cycling dynamics of the overall *Larix olgensis* plantation. We studied the decomposition and release of nutrients from stumps and coarse roots that were cleared 0, 6, 16, 26 and 33 years ago in Northeast China. The stumps and coarse roots were divided into stump discs (SD), stump knots (SK), coarse roots (>10 cm in diameter) (CR1), medium-coarse roots (5–10 cm in diameter) (CR2) and fine-coarse roots (2–5 cm in diameter) (CR3). During the entire 33-year study period, SK, CR1, CR2 and CR3 lost 87.37%, 96.24%, 75.76% and 91.98% of their initial mass, respectively. The average annual decomposition rate (k) was 0.068 for SD, 0.052 for SK, 0.092 for CR1, 0.068 for CR2 and 0.066 for CR3. After 33 years of decomposition, CR3 lost 5% of its initial C, CR2 lost 2%, and SK accumulated 1%, indicating slow C release. The N residues in SK, CR1, CR2 and CR3 were 186%, 109%, 158% and 170%, respectively. Coarse roots released P significantly faster than SD and SK, with 13% of the initial P released in CR1. SD and SK release cellulose, hemicellulose and lignin faster than coarse roots. The results show that *Larix olgensis* stumps and coarse roots could contribute to soil fertility recovery and serve as a long-term nutrient reservoir for forest vegetation.

**Keywords:** forest plantations; root biomass; below-ground biomass; decomposition

## 1. Introduction

Coarse wood debris (CWD), such as fallen wood, snag, stumps and coarse roots, plays an important role in maintaining nutrient cycling in forests. In addition, CWD provides living space for many soil organisms [1–3]. CWD is not only beneficial for maintaining biodiversity, promoting soil formation and providing a source of nutrients for subtropical plant communities but also serves as a long-term carbon and nutrient pool and an important component of carbon balance in forest ecosystems [4–7]. However, in plantations of Northeast China, other CWD (large branches, fallen wood, etc.) are transported or crushed during the management process. Therefore, these CWD mainly comprise stumps and coarse roots [8,9].

There is no consensus on whether the decomposition of tree stumps and coarse roots can provide nutrients for woodland soil. Fahey et al. (1988) suggested that the excavation and transport of stumps and coarse roots should be delayed by 3–6 months, a delay that would allow some nutrients from stumps and coarse roots to penetrate into the soil [10]. In addition, Conn (1997) and Palviainen et al. (2015) found that stumps and coarse roots remain in plantations for a long time, providing nutrient sources for forest vegetation and serving as long-term carbon pools [11,12]. However, in low-nutrient forest soils, Augusto et al. (2015) recommended that the underground biomass near tree stumps be

harvested only once per stand rotation (possibly during the felling phase) [13]. Egnell et al. (2007) considered the removal of stumps and coarse roots to not cause the severe depletion of soil nutrient stocks in mature forests due to the relatively low concentrations of nutrients in stumps and coarse roots [14]. Hakkila (2004) and Saarinen et al. (2006) also proposed that stumps and coarse roots with stump discs of a <15 cm diameter should not be harvested because they contain higher levels of nutrients, and stumps and coarse roots with larger diameters should be removed [15,16].

Carbon (C), nitrogen (N) and phosphorus (P) are three mineral nutrients that are essential for plant growth, and all of these elements are integral to the nutrient cycling and sustainable management of forest ecosystems [17]. Under the premise that large areas of forest land cannot be fertilized, nutrients that return to the soil from stumps and coarse roots play an important role in maintaining soil fertility. The nutrients and chemical compounds in different plant organs are not the same [12,18], and nutrient concentrations in coarse roots, fine roots (<2 mm in diameter) and bark vary greatly [19,20]. Nutrient transport and storage patterns differ among plant organs, where N and P are active [12]. There are few reports on the decomposition characteristics of the stumps and coarse roots of *Larix olgensis*. Thus far, studies on the decomposition of stumps and coarse roots of *Larix olgensis* have been conducted in natural forests, and the age of stump systems remains unknown [21]. However, compared with natural forests, the environments of plantation forests differ with regard to the decomposition and nutrient release in stumps and coarse roots [22].

Larch (*Larix* spp.) is the main afforestation tree species in Northeast China. It was planted there in the 1950s and 1960s and has the advantages of cold resistance and fast growth [23]. The area of larch plantation accounts for 65.10% (55.67 million ha) of the total forest reserve area in Heilongjiang Province. The trunk volume accounts for 70.13% (7766.64 million m$^3$) of the total trunk volume of forest reserves in Heilongjiang Province [24]. After nearly 30 years of renewal, the larch plantation now has a large number of larch stumps and coarse roots.

The aim of this study was (1) to determine the general law for the decomposition of *Larix olgensis* stumps and coarse roots in a chronosequence of stands that were clear-cut 0, 6, 16, 26 and 33 years ago. We also aimed (2) to determine whether the decomposition rate of nutrient dynamics (C, N and P), and chemical compounds (cellulose, hemicellulose and lignin) differ between different component of stumps and coarse roots. Therefore, we hypothesize that nutrients and chemical components are released faster from coarse roots via decomposition than stumps because of their smaller diameter closer contact with the soil. Additionally, stable moisture and temperature conditions in the soil increase the decomposition of coarse roots.

## 2. Materials and Methods

### 2.1. Site Description and Sampling

The study site was Mengjiagang Forest Farm (130°32′42″–130°52′36″ E, 46°20′16″–46°30′50″ N), Jiamusi City, Heilongjiang Province, Northeast China, where the harvesting history of the stands is well documented (Figure 1). The region has a continental monsoon climate in East Asia, with elevation ranging from 168 to 575 m. The average annual temperature is 2.7 °C. The extreme maximum temperature was 35.6 °C, and the minimum temperature was −34.7 °C. The forest stands are all *Larix olgensis* plantations before clear-cutting. More site and soil information are shown in Table 1.

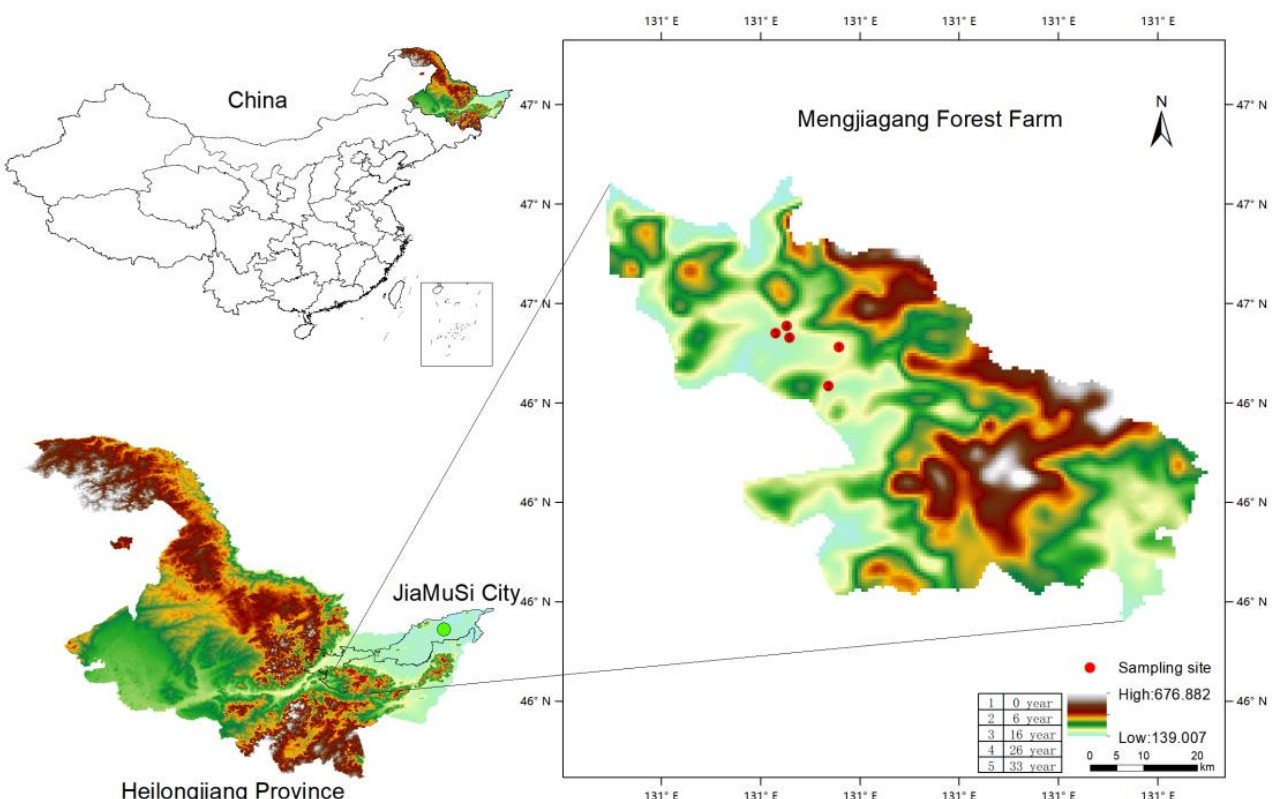

**Figure 1.** The sampling sites distribution of *Larix olgensis* stumps and coarse roots.

**Table 1.** Size, average diameter at stump disc and forest management practices in the study stands clear-cut 0, 6, 16, 26 and 33 years prior to sampling.

| Time (Years) | Size of the Stand (ha) | Stand Density (Trees·ha⁻¹) | Mean Tree Height (m) | Soil Type | Exposure | Slope (°) | Slope Position | DBH (cm) | D (cm) | Intermediate Cutting Times |
|---|---|---|---|---|---|---|---|---|---|---|
| 0 | 0.04 | 661 | 24.30 | dark brown soil | southwest | 9 | mid slope | 24.60 | 27.97 | 2 |
| 6 | 0.04 | 2860 | 3.93 | dark brown soil | east | 11 | mid slope | 3.50 | 33.64 | - |
| 16 | 0.04 | 3357 | 9.56 | dark brown soil | west | 8 | mid slope | 8.85 | 31.04 | - |
| 26 | 0.04 | 1675 | 15.86 | dark brown soil | east | 13 | mid slope | 13.60 | 24.34 | 1 |
| 33 | 0.04 | 1525 | 20.43 | Albic soil | southwest | 10 | mid slope | 16.22 | 17.43 | 1 |

Note: (*D*): Average diameter at stump disc; (*DBH*): mean diameter at breast high.

*Larix olgensis* stumps and coarse roots were collected over 6 days in August 2009 from stands that were clear-cut 0, 6, 16, 26 and 33 years ago. In each stand, three 10 m long transects were laid out in the center of stands, and samples from one stump and coarse roots were collected from each transect. The distance between transects was 10 m. The average diameters of the studied stump disc were 27.97, 33.64, 31.04, 24.34 and 17.43 cm at 0, 6, 16, 26 and 33-year-old sites, respectively. The diameter ranges for the stump discs at the 0, 6-, 16-, 26- and 33-years old sites were 12.3–47.1 cm, 14.4–47.1 cm, 11.4–63.2 cm, 11.5–41.6 cm and 5.0–35.9 cm, respectively.

The stumps and coarse roots were manually excavated with shovels, and the uprooting procedure was repeated until the complete stumps and coarse roots had been removed. The stumps and coarse roots were cut and weighed according to their components and sorted into stump disc (SD), stump knot (SK), coarse roots (>10 cm in diameter) (CR1), medium-coarse roots (5–10 cm) (CR2) and fine-coarse roots (2–5 cm) (CR3) (Figure 2).

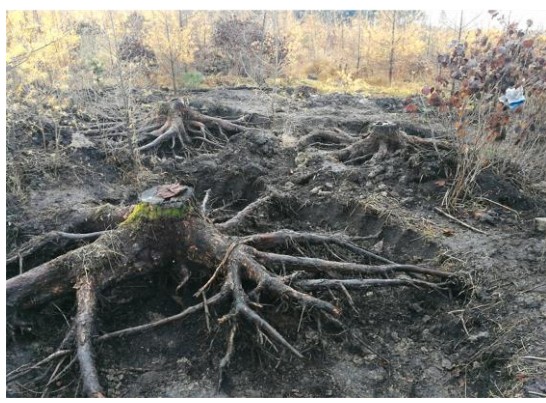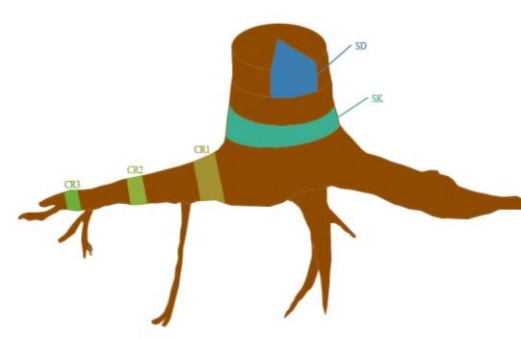

**Figure 2.** Sampling locations in decomposing *Larix olgensis* stumps and coarse roots.

### 2.2. Laboratory Analyses

For each sampled stump disc (including wood and bark), diameter (10 cm above the root collar) [25] and fresh weight were recorded. A 5 cm disc subsample was cut from the middle of each stump disc, and fresh weights were recorded. Similarly, the fresh weights of the stump knot and a 5 cm disc subsample of each stump knot were recorded.

Subsamples of approximately 100 g of coarse roots were randomly sampled to determine their exact fresh weights. All the subsamples were oven-dried at 85 °C until a constant weight, and then the ratio of dry to fresh weight was calculated. The dry biomass of each portion of stumps and coarse roots was calculated by multiplying their fresh weight by the respective dry/fresh weight ratio [13].

The volumes of the samples were gravimetrically determined using the water displacement method [26]. The samples were milled, and the concentrations of C and N were determined using an elemental combustion analyzer (The Elemental Combustion System 4024, Bussero, Italy), and P concentration was determined with the phospho-vanado-molybdate colorimetric method, following digestion with $H_2SO_4$-$HClO_4$ [27]. The concentrations of cellulose, hemicellulose and lignin were determined using an automatic fiber analyzer (ANKOM A2000i, Macedon, NY, USA) [28].

### 2.3. Calculations and Statistical Analyses

Bulk densities ($\rho$, in g/cm$^3$) of the stumps and coarse roots samples were calculated by the following equation:

$$\rho = \frac{m}{v} \tag{1}$$

where $m$ is the dry mass of the sample, and $v$ is the volume of the sample. Masses of the stumps and coarse roots ($p$) as a percentage of their initial masses were calculated with the following equation:

$$p = \frac{m_t}{m_0} \times 100 \tag{2}$$

where $m_0$ is the initial mass and $m_t$ is the mass at time of sampling. The initial mass of the stumps and coarse roots was calculated by using the average bulk density of samples collected immediately after clear cutting. The decomposition rate constant $k$ (year$^{-1}$) was calculated for the different decomposition periods with the following formula:

$$k = \ln(m_0/m_t)/t \tag{3}$$

where $m_0$ is the initial dry mass; $m_t$ is the dry mass at time of sampling; and $t$ is the length of the decomposition period in years. Density, nitrogen, cellulose and hemicellulose of the

stumps and coarse roots over time were described by the negative exponential function, which is commonly used for describing the decomposition rate of CWD:

$$y = a\mathrm{e}^{-bt} \tag{4}$$

where $t$ is time since harvesting in years, and $a$ and $b$ are parameters.

Differences in mass loss and density, decomposition rate constants ($k$), cellulose, hemicellulose, lignin, the concentrations of nutrients and C/N ratios among stumps and coarse roots factions (SD, SK, CR1, CR2 and CR3) were analyzed with a mixed linear model followed by a Bonferroni test. Stumps and coarse roots fractions, the length of decomposition period and their interactions were defined as fixed factors. Differences were considered statistically significant when $p$ was $\leq 0.05$.

## 3. Results

### 3.1. Decomposition

The decomposition period and the fraction of the stumps and coarse roots are important factors for explaining density, mass and C losses, as well as differences in the decomposition rate constant ($k$) (Table 2). The initial densities of SD (0.521 g/cm$^3$), SK (0.488 g/cm$^3$), CR1 (0.543 g/cm$^3$), CR2 (0.557 g/cm$^3$) and CR3 (0.542 g/cm$^3$) were similar (Figure 2). From the 6th year, the density of CR3 was significantly lower than that of other components. Furthermore, 33 years after decomposition, the density of SK (0.203 g/cm$^3$) was lower than that of other components, and the density of CR1 (0.388 g/cm$^3$) was the highest, being significantly higher than that of other components ($p \leq 0.05$).

**Table 2.** The results of the mixed model analysis for the fixed factors explaining the changes in density, mass loss, carbon loss and $k$-values of stumps and coarse roots.

| Factor | Degrees of Freedom | *F*-Value | Significance, *p*-Value |
|---|---|---|---|
| Density | | | |
| Intercept | 1 | 4853.04 | <0.0001 |
| Decomposition period | 3 | 35.20 | <0.0001 |
| Fraction | 1 | 284.83 | <0.0001 |
| Fraction × decomposition period | 3 | 1.98 | 0.170 |
| Mass loss | | | |
| Intercept | 1 | 284.81 | <0.0001 |
| Decomposition period | 3 | 5.30 | 0.015 |
| Fraction | 1 | 132.81 | <0.0001 |
| Fraction × decomposition period | 3 | 10.75 | <0.001 |
| C loss | | | |
| Intercept | 1 | 184.91 | <0.0001 |
| Decomposition period | 3 | 38.05 | <0.0001 |
| Fraction | 1 | 47.09 | <0.0001 |
| Fraction × decomposition period | 3 | 12.81 | <0.0001 |
| k-value | | | |
| Intercept | 1 | 1834.35 | <0.0001 |
| Decomposition period | 3 | 44.32 | <0.0001 |
| Fraction | 1 | 461.10 | <0.0001 |
| Fraction × decomposition period | 3 | 8.93 | <0.001 |

The mass loss of each component was different at different stages of decomposition. SD mass loss was 100% after 33 years of decomposition (Figure 3). SK, CR1, CR2 and CR3 lost 87.37%, 96.24%, 75.76% and 91.98% of their initial mass in 33 years, respectively, and the difference between CR1 and CR2 was statistically significant ($p \leq 0.05$). At 16 years of decomposition, all components except CR2 (125%) were in a C release state. At the 26th year of decomposition, all components except CR3 were in a C accumulation state. Over 33 years, CR3 released carbon at a significantly faster rate than the other components; it

lost 5% of its initial C, 4% of CR1, 2% of CR2 and 1% of accumulated SK. The negative exponential model can accurately describe the decomposition of stumps and coarse roots (density, nitrogen, cellulose and hemicellulose), as well as calculating the determination coefficients of each component range from 0.34 to 0.78, 0.38 to 0.90, 0.30 to 0.67 and 0.29 to 0.70 (Table 3).

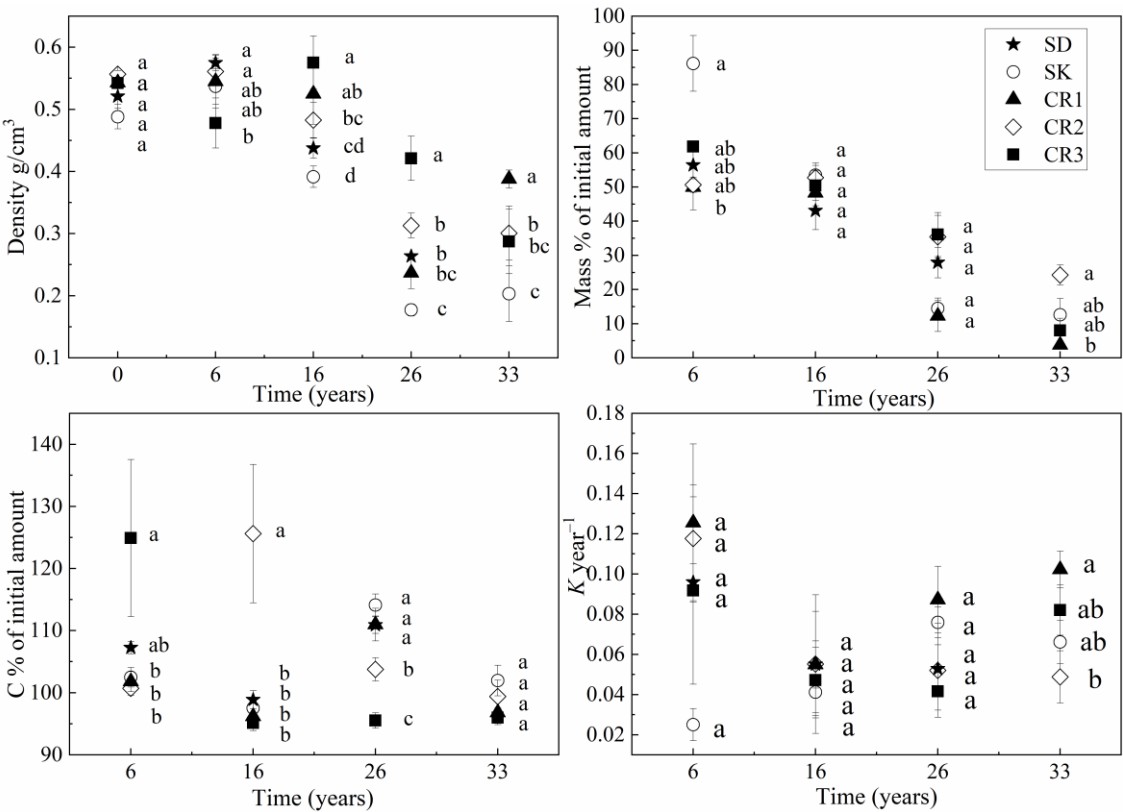

**Figure 3.** Mean (±standard error) density, mass and carbon loss and decomposition rate constants, *k* (Equation (3)) of SD, SK, CR1, CR2 and CR3 after various periods of decomposition. Different letters indicate statistically significant differences ($p \leq 0.05$) between SD, SK, CR1, CR2 and CR3.

**Table 3.** The number of observations, the estimates, standard errors of parameters and the adjusted $R^2$ values of negative exponential ($y = ae^{-bt}$) for density, nitrogen, cellulose and hemicellulose as a function of time (*t*, years) in China *Larix olgensis* stumps and coarse roots.

| Dependent Variable | *n* | Parameter a | | | Parameter b | | | Adj.$R^2$ |
|---|---|---|---|---|---|---|---|---|
| | | Value | SE | *p* | Value | SE | *p* | 0.499 |
| Density | | | | | | | | |
| SD | 15 | 0.578 | 0.003 | <0.0001 | 0.022 | 0.005 | <0.001 | 0.693 |
| SK | 15 | 0.550 | 0.034 | <0.0001 | 0.030 | 0.004 | <0.0001 | 0.763 |
| CR1 | 15 | 0.576 | 0.045 | <0.0001 | 0.017 | 0.005 | <0.001 | 0.428 |
| CR2 | 15 | 0.594 | 0.027 | <0.0001 | 0.020 | 0.003 | <0.0001 | 0.775 |
| CR3 | 15 | 0.563 | 0.042 | <0.0001 | 0.013 | 0.004 | <0.01 | 0.338 |
| nitrogen | | | | | | | | |
| SD | 15 | 1.513 | 0.155 | <0.0001 | −0.047 | 0.005 | <0.0001 | 0.904 |
| SK | 15 | 1.054 | 0.175 | <0.0001 | −0.032 | 0.006 | <0.0001 | 0.637 |
| CR1 | 15 | 1.220 | 0.139 | <0.0001 | −0.010 | 0.005 | <0.001 | 0.427 |
| CR2 | 15 | 1.503 | 0.196 | <0.0001 | −0.019 | 0.005 | <0.001 | 0.382 |
| CR3 | 15 | 1.360 | 0.175 | <0.0001 | −0.028 | 0.005 | <0.0001 | 0.665 |

**Table 3.** *Cont.*

| Dependent Variable | n | Parameter a | | | Parameter b | | | Adj.$R^2$ 0.499 |
|---|---|---|---|---|---|---|---|---|
| | | Value | SE | p | Value | SE | p | |
| Cellulose | | | | | | | | |
| SD | 15 | 498.900 | 46.040 | <0.0001 | 0.041 | 0.010 | <0.001 | 0.669 |
| SK | 15 | 454.400 | 36.700 | <0.0001 | 0.023 | 0.006 | <0.01 | 0.546 |
| CR1 | 15 | 395.200 | 27.410 | <0.0001 | 0.004 | 0.004 | <0.01 | 0.263 |
| CR2 | 15 | 394.700 | 26.640 | <0.0001 | 0.002 | 0.003 | <0.01 | 0.307 |
| CR3 | 15 | 444.200 | 24.980 | <0.0001 | 0.0093 | 0.003 | <0.01 | 0.296 |
| Hemicellulose | | | | | | | | |
| SD | 15 | 123.792 | 11.596 | <0.0001 | 0.047 | 0.011 | <0.001 | 0.704 |
| SK | 15 | 128.500 | 10.110 | <0.0001 | 0.030 | 0.006 | <0.001 | 0.659 |
| CR1 | 15 | 109.800 | 7.430 | <0.0001 | 0.011 | 0.004 | <0.01 | 0.293 |
| CR2 | 15 | 105.700 | 4.517 | <0.0001 | 0.006 | 0.002 | <0.01 | 0.294 |
| CR3 | 15 | 107.000 | 2.598 | <0.0001 | 0.004 | 0.001 | <0.01 | 0.331 |

During 26 years of decomposition, the decomposition rate constants ($k$) of each component showed no significant difference (Figure 3). Over the entire 33-year study period, the average annual $k$ values of SD, SK, CR1 and CR3 were 0.068 (range 0.053 to 0.096), 0.052 (range 0.025 to 0.076), 0.092 (range 0.055 to 0.125) and 0.068 (range 0.049 to 0.118), respectively. The mean annual $k$ value of CR3 was 0.066 (range from 0.042 to 0.092). After 33 years of decomposition, the $k$ value of CR2 was significantly different from that of other components ($p \leq 0.05$).

*3.2. Changes in Nutrient Concentrations, Cellulose, Hemicellulose, Lignin and C/N Ratios*

The fractions of stumps and coarse roots, the length of decomposition period and their interactions can account for changes in nutrient concentrations, C/N ratios, cellulose, hemicellulose, lignin and lignin/cellulose ratios (Table 4). There were no significant differences in the initial carbon concentration of each component of the stumps and coarse roots ($p > 0.05$) (Figure 4). In the decomposition process, the carbon concentration varied from 1 to 4% units; the nitrogen concentration varied from 1 to 2% units; and the phosphorus concentration varied from 1 to 5 mg/kg, in which the nitrogen concentration was twice the initial value. After 33 years of decomposition, there was no significant difference in the C content of each component. The concentration of N and P was the highest in CR3, which was significantly higher than that of other components ($p < 0.05$), followed by SK and CR2, and the concentration of CR1 was significantly lower than that of other components ($p < 0.05$). The initial C/N values of CR1 and SD were significantly higher than those of CR2 and CR3 ($p < 0.05$). However, after 33 years of decomposition, the C/N value of CR1 among the components was the highest, followed by CR2, SK and CR3, with significant differences between the components ($p < 0.05$). For 0–33 years, the C/N ratio of SK decreased from 24 to 13; the C/N ratio of CR1 decreased from 28 to 25; the C/N ratio of CR2 decreased from 23 to 14; and the C/N ratio of CR3 decreased from 19 to 11.

**Table 4.** The results of the mixed model analysis for the fixed factors explaining the changes in C/N ratio and carbon, nitrogen, phosphorus, cellulose, hemicellulose and lignin of stumps and coarse roots.

| Factor | Degrees of Freedom | F-Value | Significance, p-Value |
|---|---|---|---|
| C concentration | | | |
| Intercept | 1 | 90,034.17 | <0.0001 |
| Decomposition period | 3 | 19.66 | <0.0001 |
| Fraction | 1 | 37.59 | <0.0001 |
| Fraction × decomposition period | 3 | 22.57 | <0.0001 |

**Table 4.** *Cont.*

| Factor | Degrees of Freedom | *F*-Value | Significance, *p*-Value |
|---|---|---|---|
| C/N ratio | | | |
| Intercept | 1 | 6319.73 | <0.0001 |
| Decomposition period | 3 | 399.69 | <0.0001 |
| Fraction | 1 | 22.88 | <0.0001 |
| Fraction × decomposition period | 3 | 56.17 | <0.0001 |
| N concentration | | | |
| Intercept | 1 | 1972.80 | <0.0001 |
| Decomposition period | 3 | 178.86 | <0.0001 |
| Fraction | 1 | 1.99 | 0.184 |
| Fraction × decomposition period | 3 | 16.17 | <0.0001 |
| P concentration | | | |
| Intercept | 1 | 20,986.30 | <0.0001 |
| Decomposition period | 3 | 280.95 | <0.0001 |
| Fraction | 1 | 2223.26 | <0.0001 |
| Fraction × decomposition period | 3 | 2.04 | 0.162 |
| Cellulose | | | |
| Intercept | 1 | 5683.81 | <0.0001 |
| Decomposition period | 3 | 57.35 | <0.0001 |
| Fraction | 1 | 31.60 | <0.0001 |
| Fraction × decomposition period | 3 | 2.30 | 0.130 |
| Hemicellulose | | | |
| Intercept | 1 | 91,766.13 | <0.0001 |
| Decomposition period | 3 | 389.58 | <0.0001 |
| Fraction | 1 | 1139.43 | <0.0001 |
| Fraction × decomposition period | 3 | 8.03 | 0.003 |
| Lignin | | | |
| Intercept | 1 | 2796.74 | <0.0001 |
| Decomposition period | 3 | 33.95 | <0.0001 |
| Fraction | 1 | 1.96 | 0.187 |
| Fraction × decomposition period | 3 | 10.30 | 0.001 |

The initial SD cellulose content was significantly higher than that of other components ($p < 0.05$). There were no significant differences in cellulose content between 6 and 16 years or in the initial hemicellulose content ($p > 0.05$), and the initial lignin CR1 content was significantly lower than that of other components ($p < 0.05$). However, in the 6th year of decomposition, there was no significant difference in all components of the stumps and coarse roots. In the 26th year of decomposition, the contents of cellulose, hemicellulose and lignin in SD were the lowest, and the contents of cellulose, hemicellulose and lignin in CR3 were the highest. After 33 years of decomposition, the contents of cellulose, hemicellulose and lignin in SK were lower than those of other components. There was no significant difference in the lignin/cellulose content from the initial to the 6th year of decomposition ($p > 0.05$), but the lignin/cellulose values of SD and SK at the 26th year of decomposition were significantly higher than that of coarse roots ($p < 0.05$). After 33 years of decomposition, there were no significant differences in the contents of each component.

### 3.3. Nutrient Release

The decomposition period length and fraction are significant factors explaining N release (Table 5). After 33 years of decomposition, the residues of N in SK, CR1, CR2 and CR3 reached 186%, 109%, 158% and 170%, respectively (Figure 5). The P release rate of coarse roots was significantly faster than that of SD and SK ($p < 0.05$). P content in 33 years of decomposition of SK is higher than the initial amount (171%). Instead, 13% of the initial P is released from CR1. SD and SK release cellulose, hemicellulose and lignin faster than coarse roots. At the 26th year of decomposition, the release of each component

was significant, and 87%, 93% and 77% of the initial cellulose, hemicellulose and lignin contents were lost in SD, respectively. SK lost 71%, 78% and 60% of the initial cellulose, hemicellulose and lignin contents, respectively. After 33 years of decomposition, there was no significant difference between the levels of cellulose, hemicellulose and lignin residues among the three coarse roots components, and SK released 38%, 49% and 16% of the initial cellulose, hemicellulose and lignin contents.

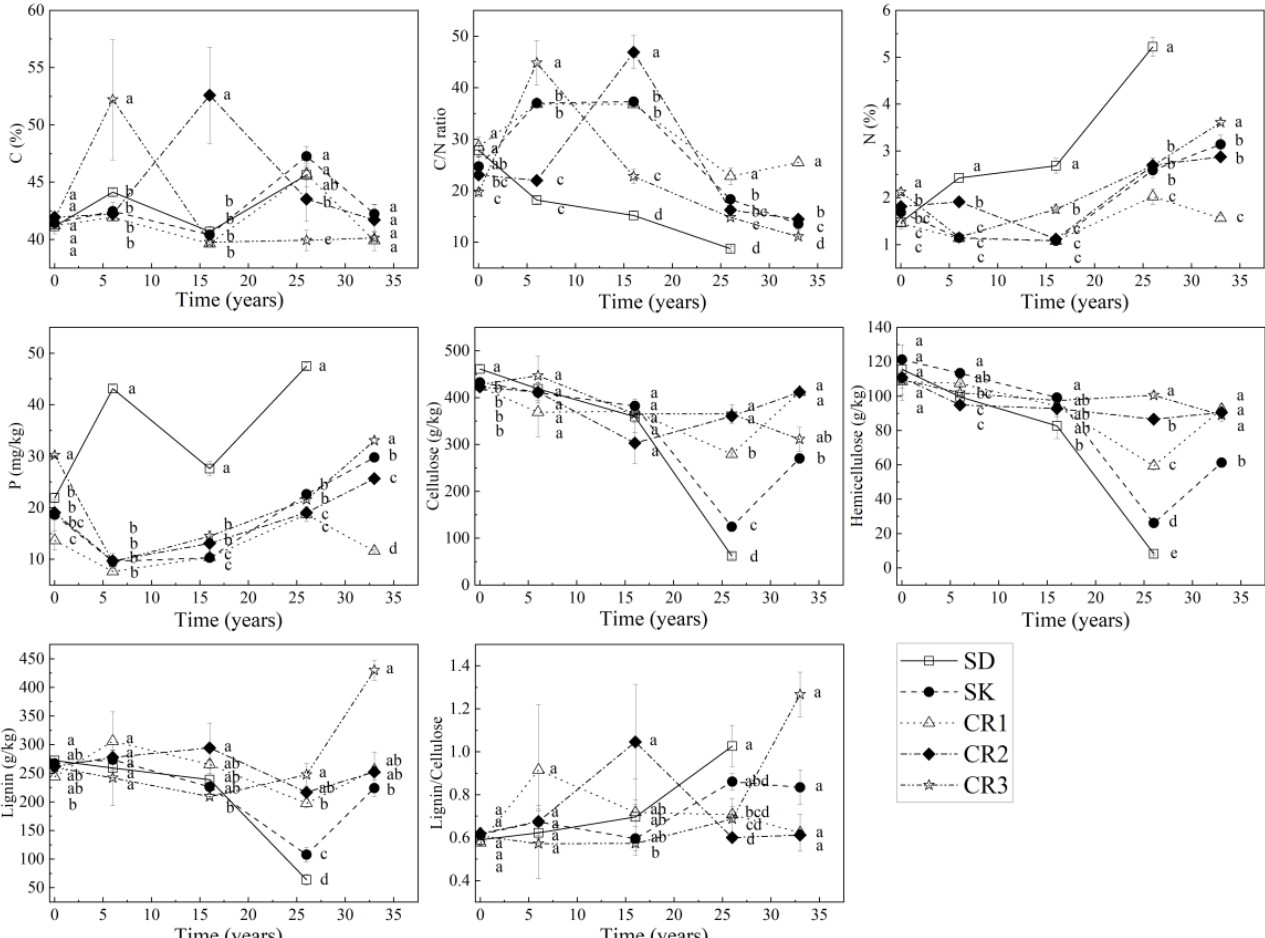

**Figure 4.** Mean (±standard error) carbon, nitrogen, phosphorus, cellulose, hemicellulose and lignin concentrations and C/N, lignin/cellulose ratios in SD, SK, CR1, CR2 and CR3 after various periods of decomposition. Different letters indicate statistically significant differences ($p \leq 0.05$) between SD, SK, CR1, CR2 and CR3.

**Table 5.** The results of the mixed model analysis for the fixed factors explaining the carbon, nitrogen, phosphorus, cellulose, hemicellulose and lignin release from stumps and coarse roots.

| Factor | Degrees of Freedom | *F*-Value | Significance, *p*-Value |
|---|---|---|---|
| N release | | | |
| Intercept | 1 | 214.45 | <0.0001 |
| Decomposition period | 3 | 17.25 | <0.0001 |
| Fraction | 1 | 33.33 | <0.0001 |
| Fraction × decomposition period | 3 | 17.59 | <0.0001 |
| P release | | | |
| Intercept | 1 | 217.76 | <0.0001 |
| Decomposition period | 3 | 15.31 | <0.0001 |
| Fraction | 1 | 15.73 | 0.002 |
| Fraction × decomposition period | 3 | 4.56 | 0.024 |

**Table 5.** *Cont.*

| Factor | Degrees of Freedom | *F*-Value | Significance, *p*-Value |
|---|---|---|---|
| Cellulose release | | | |
| Intercept | 1 | 161.27 | <0.0001 |
| Decomposition period | 3 | 8.91 | 0.001 |
| Fraction | 1 | 0.11 | 0.746 |
| Fraction × decomposition period | 3 | 2.07 | 0.148 |
| Hemicellulose release | | | |
| Intercept | 1 | 88.30 | 0.0001 |
| Decomposition period | 3 | 0.30 | 0.825 |
| Fraction | 1 | 11.79 | 0.005 |
| Fraction × decomposition period | 3 | 0.82 | 0.506 |
| Lignin release | | | |
| Intercept | 1 | 65.46 | <0.0001 |
| Decomposition period | 3 | 17.24 | <0.0001 |
| Fraction | 1 | 0.67 | 0.426 |
| Fraction × decomposition period | 3 | 8.69 | 0.001 |

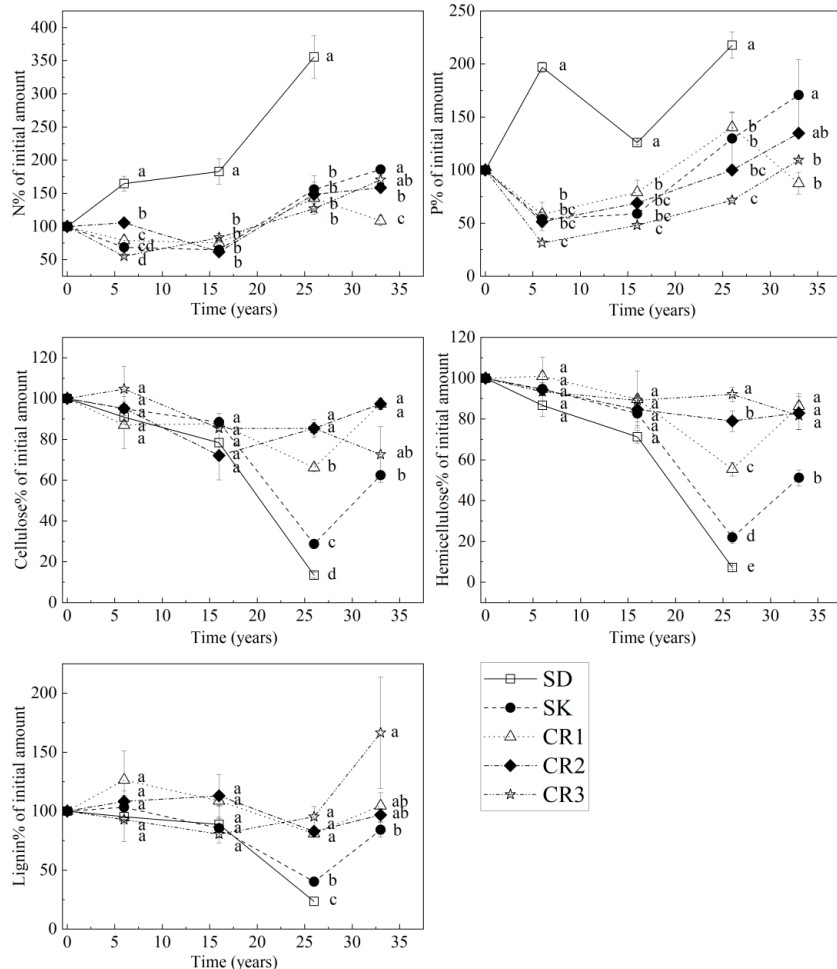

**Figure 5.** The amount of nitrogen, phosphorus, cellulose, hemicellulose and lignin as a percentage of the initial amount in SD, SK, CR1, CR2 and CR3 after various periods of decomposition. Different letters indicate statistically significant differences (*p* < 0.05) between SD, SK, CR1, CR2 and CR3.

The decomposition ratio (*Fm*) of SD (0.82) was significantly higher than that of SK, CR1 and CR3 (*p* < 0.05) (Table 6). The decomposition ratios of CR2 and CR3 were 0.73

and 0.61, respectively, second only to SD. The decomposition ratio of CR1 (0.51) was the smallest but was not significant for SK.

**Table 6.** The ratio of each component of the stumps and coarse roots to be decomposed easily.

| Component | SD | SK | CR1 | CR2 | CR3 |
|---|---|---|---|---|---|
| *Fm* | $0.82 \pm 0.03$ a | $0.56 \pm 0.05$ bc | $0.51 \pm 0.14$ c | $0.73 \pm 0.01$ ab | $0.61 \pm 0.14$ bc |

Note: CENTURY model [29]. Different letters indicate statistically significant differences ($p < 0.05$) between SD, SK, CR1, CR2 and CR3.

## 4. Discussion

The decomposition rate, density and mass loss of stumps are greater than those for coarse roots, which is not consistent with the initial hypothesis. The reason for this phenomenon is that the cut surfaces of stumps are exposed to the soil surface, which is not only warmer than the deeper soil but also subject to rainwater leaching [12]. In addition, the activity of animals (mostly invertebrates) at the surface of the soil breaks up the stumps and, thus, accelerates decomposition by introducing a large number of bacteria and fungi that promote nutrient cycling [30,31]. In addition, coarse roots are entirely covered by bark in the early stages of decomposition, keeping them drier and forming a barrier for microbial colonization, thereby slowing down decomposition [32,33]. The density is opposite to the change in mass loss rate, which means that the smaller the density is, the easier it is for the stumps and coarse roots to decompose, and the higher the mass loss rate is [34]. With the higher degree of decomposition, the density of each component of the stumps and coarse roots gradually decreases and maintains a stable trend in the late decomposition stage. This is because after the nutrients (such as N, P, and K) are decomposed by microorganisms, chemical compounds (such as cellulose, lignin, tannin, etc.) gradually increase, leading to slow decomposition [35].

The average annual decomposition rate constant (k) of stumps and coarse roots was higher than that of CWD in a larch plantation (0.019 (0.009–0.037)) [36]. In addition, the *k* value of stumps and coarse roots (0.071) is also higher than the average annual decomposition rate constant of fallen larch wood (0.0136) [37]. However, the stumps and coarse roots k was lower than the annual decomposition rate constants of larix fine roots (5–10 mm (0.1967), 2–5 mm (0.0955), 2 mm (0.2464)) and larch litter (0.225) [38]. The negative exponential model can better describe the changes in density, nitrogen, cellulose and hemicellulose during the decomposition of stumps and coarse roots. For carbon and lignin, due to the different decomposition patterns and the existence of initial lag phase, a negative exponential model is not applicable [39,40]. In terms of predicting the three stages of CWD decomposition, the decomposition is slow for the first 5–10 years, followed by rapid decomposition for 10 years, after which decomposition slows down once more [41]. Evidently, the decomposition of the stumps and coarse roots takes longer. Decomposition is slow in the first 16 years, fast between 16 and 26 years and then tends to slow down.

C, N and P are essential nutrients in plant growth and are crucial for nutrient cycling and sustainable management in forest ecosystems [17]. However, different anatomical structures of wood have different physical structures and chemical compounds, and the content of compounds and elements varies greatly [42,43]. The carbon content of stumps and coarse roots is not the same in each period, and the average carbon content is less than 50%, indicating that CWD cannot be accurately estimated using a single mean value [44]. The decomposition time for stumps and coarse roots residues was longer, and the carbon loss rate was not synchronous with the decomposition rate of wood. Therefore, C content increased in the 26th year of decomposition. In addition, the difference in the carbon contents of different components was significantly affected by lignin content [45], and when lignin accumulated, the release of carbon content was low (33 years).

The C/N ratio can be used to characterize the mass change and nutrient limitation status of a substrate during CWD decomposition and can also indicate the direction of material cycling and energy flow [46–48]. Ecological stoichiometry theory suggests that in

most cases, C/N > 27, microbial growth is limited [49], and thus, stumps and coarse roots contained limited amounts of N between 6 and 16 years of decomposition.

The accumulation of N and P in the stumps and coarse roots occurred after 16 years of decomposition. Manzoni et al. (2010) consider that with fragmentation of the stump, the leaching of rainwater and the rapid decomposition of organic matter causes P accumulation [50]. Another idea is that the hyphae of decaying bacteria are capable of transferring P from the soil to the stumps systems [51]. Coarse roots release N faster than SD. This is consistent with the results of Wei and Chen et al., who showed that coarse roots have a higher N fixation rate than stumps [52,53].

Cellulose, hemicellulose and lignin essentially show a trend of release and then accumulation. After 26 years of decomposition, cellulose, hemicellulose and lignin release from coarse roots (CR1, CR2, CR3) was significantly slower than the release of these compounds from SD and SK. This is related to the increase in N content during the decomposition of stumps and coarse roots. A high N content can promote the decomposition of lignin by microorganisms (mainly basidiomycetes) [54]. However, during CWD decomposition, lignin and cellulose are not independently degraded, therefore, and other substances are protected by lignin [55], indirectly promoting the decomposition of cellulose and hemicellulose. During 26–33 years of decomposition, the contents of three compounds (cellulose, hemicellulose and lignin) increased and began to accumulate. This was related to the "S" growth of fungi during the decomposition of woody residues [56]. In the later stages of decomposition, the number of white rot fungi increased, and decomposition accelerated, resulting in accumulation [57]. In addition, it was also found that lignin analogues can be formed by microbial metabolism during decomposition [34], which significantly enriches lignin. The decomposition index (*Fm*) can reflect the decomposition ability of plant residues to a certain extent using the initial contents of lignin and N in plant bodies [58]. SD is more likely to decompose due to its high N concentration and low C/N ratio, and low C/N was more amenable to microbial decomposition [49].

When determining volume density in water, losses due to fragmentation are not taken into account, and measurements may become less accurate later in the decomposition process [39]. The time series approach assumes that factors other than time are similar at all study sites [5]. Initial conditions and thinning times between sites will affect the results to some extent [59]. For this reason, we chose study sites that were not far apart and had similar climate, vegetation and soil types, meaning that differences were less likely to occur. Root decomposition is a complex ecological process affected by climate and root characteristics [60]. Nutrient dynamics are only superficial phenomena in the decomposition process of stumps and coarse roots. Therefore, environmental factors and microbial dynamics should be combined in future research on the decomposition process of stumps and coarse roots.

## 5. Conclusions

The decomposition of stumps and coarse roots is a long-term process, generally divided into two stages: 0–16 years of slow decomposition; 16–33 years of gradually accelerated decomposition. The general law of decomposition is to proceed from the outside in, starting with a stump disc (SD) and coarse roots with diameters of 2–5 cm (CR3). There are different laws governing the release of nutrients and chemical compounds in the decomposition processes of stumps and coarse roots. Stumps decompose and release chemical compounds (cellulose, hemicellulose and lignin) at a faster rate than coarse roots but release C, N and P more slowly than coarse roots. In summary, Larch stumps and coarse roots exist for a long time after clear-cutting and can provide long-term nutrient sources for soil and vegetation in plantations.

**Author Contributions:** Conceptualization, X.M., X.C. and X.W.; investigation, X.M. and Y.Y.; methodology, X.M., Y.Y. and X.C.; data curation and formal analysis, X.M. and X.W.; software and visualization, X.M.; writing—original draft, X.M.; writing—review and editing, X.C.; funding acquisition, X.C. All authors have read and agreed to the published version of the manuscript.

**Funding:** This work was supported by the National Natural Science Foundation of China (NSFC, grant number 31870612).

**Data Availability Statement:** The data are available on reasonable request.

**Acknowledgments:** We gratefully acknowledge the assistance of numerous staff from Meng Jia Gang Forest farm with the field investigation and sample processing.

**Conflicts of Interest:** The authors declare no conflict of interest.

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
