# Peer review of "Decomposition and Nutrient Release from Larix olgensis Stumps and Coarse Roots in Northeast China 33-Year Chronosequence Study"

_forests, doi:10.3390/f14061253_

Round 1

Reviewer 1 Report

Dear Authors,

although the manuscript forests-2451487 presents an original idea and a lot of work done, it suffers from several criticalities, which need to be addressed before its possible publication in Forests.

In particular,

·        The decomposition of the different parts of the plant depends not only on the different characteristics of the different matrices, but also on the chemico-physical and biological characteristics of the sites in which they decompose. Nothing is said about the characteristics of the study area: how far apart are the stands of different ages, have they the same climatic characteristics, the same slope and exposure, what are the characteristics of the soils? Only at the end of the Discussion section (L. 353-354) is it said that the characteristics of the study sites are similar, but they have never been reported.

·        If an increase in respect to the initial value is foreseeable for N and P, how can the Authors explain the increase of cellulose, hemicellulose and lignin in several matrices at the final time with respect to the previous one?

Moreover, several minor suggestions are provided to improve the whole manuscript. In particular,

·        Use the italic for the species name.

·        Several claims seem strange and not adequately justified: soil structure (L. 30); energy source (L. 32-33); P enrichment (L. 323-324); N-rich environment promoting N fixation (L. 339-340).

·        Several statements are incorrect: C as “mineral nutrient” (L. 301); organic matter as “nutrient element” and these are indecomposable (L. 282-285).

·        Several paragraphs are vague or unclear: L. 53-60; L. 35 and 61; L. 276-277; L. 315; L. 325-328; L. 336-336. Particularly the aim of the study (L. 68-76) should be clarified: please, specify the goal of the research before describing the employed approach.

·        Substitute “chemical components” with “chemical compounds” and “altitude” with “elevation”.

·        Regarding the Material and Methods section, the employed methods are poorly described, lacking of important information or appropriate citations (L. 112-118). In addition, please indicate the ranges of the elevation and temperature (L. 83-84), as well as of the disc diameter (L. 90), rather than the mean values.

·        In Table 1, please add information on forest soils.

English should be revised.

Reviewer 2 Report

I have read your paper carefully. Generally this paper is well documented. However I have some suggestions. 

1- Some of below sentences are unclear. Please sprese these sentences during revision.

* The role of stumps and coarse roots decomposition in providing long-term nutrients  for forest land is controversial. Some studies have suggested that stumps and coarse roots  as a long-term carbon pool and nutrient source for boreal forest vegetation [10,11]. Fahey  et al. (1988) also recommends delaying stumps and coarse roots extraction and transport 40 for 3–6 months after stem cutting and this delay will allow some nutrients from dead roots  to leach into the soil [12].

* The study stands were randomly selected from forests clear-cut 0, 6, 16, 26 and 33 years  ago in the Meng Jiagang experimental forests (130°32′42″–130°52′36″ E, 46°20′16″– 80 46°30′50″N) of the Jiamusi City in Northeast China's Heilongjiang Province where the har- 81 vesting history of the stands is well documented.

2- Author mentioned in many places regarding C, N and P cycling and their crucial role both soil producivity and forests. However, this issues are needed to extend in the introduction section.

3- Discussion;

 Starting expressions  should not start like start. So, These sentences are needed rephrases; 

"Contrary to the initial assumption, SD and SK decompose faster than coarse roots.  Compared with coarse roots, the density and mass loss of SD and SK are faster. The reason  why SD is decomposed completely first is that it is exposed to the soil surface and the  temperature is higher than that in the deep soil, which is conducive to the activities of soil animals, among which invertebrates play an important role"

4- Conclusion is like results part. So, It is not good. Please rewrite conclusion part.

Best Regards

I think this paper is needed detailed english editing because some of sentences are unclear becaue of poorly academic english. 

Author Response

请看附件。

Round 2

Reviewer 1 Report

Dear Authors,

The new version of the ms forests-2451487 has been sufficiently improved over the former version. Only the caption of the Table 1 should be amended according to the new information added, but it can be done also along with the proof corrections.